# Deciphering the Impact of *HER2* Alterations on Non-Small-Cell Lung Cancer: From Biological Mechanisms to Therapeutic Approaches

**DOI:** 10.3390/jpm12101651

**Published:** 2022-10-04

**Authors:** Christophe Bontoux, Jonathan Benzaquen, Véronique Hofman, Simon Heeke, Paul Hannetel, Pierre Capela-Brosseau-Laborde, Charles-Hugo Marquette, Marius Ilié, Paul Hofman

**Affiliations:** 1Laboratory of Clinical and Experimental Pathology, Université Côte d’Azur, Pasteur Hospital, Centre Hospitalier Universitaire de Nice, Biobank BB-0033-00025, 06000 Nice, France; 2IRCAN Team 4, Inserm U1081/CNRS 7284, Centre de Lutte Contre le Cancer Antoine Lacassagne, 06000 Nice, France; 3FHU OncoAge, Centre Hospitalier Universitaire de Nice, 06000 Nice, France; 4Department of Pneumology, Pasteur Hospital, University Côte d’Azur, 30 Avenue de la Voie Romaine, 06000 Nice, France; 5Department of Thoracic/Head & Neck Medical Oncology, The University of Texas, MD Anderson Cancer Center, Houston, TX 77030, USA

**Keywords:** non-small-cell lung cancer, lung cancer, *HER2*, mutations, amplification, targeted therapies, monoclonal antibody, tyrosine kinase inhibitor, molecular pathology, next-generation sequencing

## Abstract

Despite the recent increase in the number of types of treatments, non-small-cell lung cancer (NSCLC) remains the major cause of death from cancer worldwide. So, there is an urgent need to develop new therapeutic strategies. The *HER2* gene codes for tyrosine kinase receptor whose alterations are known to drive carcinogenesis. *HER2* alterations, including amplification, mutations, and overexpression, have been mainly described in breast and gastric cancers, but up to 4% of NSCLC harbor actionable *HER2* mutations. HER2-targeted therapy for NSCLC with trastuzumab, pertuzumab, and trastuzumab emtansine has failed to demonstrate an improvement in survival. Nevertheless, recent data from phase II trials have shed light on promising specific therapies for *HER2*-mutant NSCLC such as trastuzumab deruxtecan. Herein, we aimed to provide an updated review on the biology, epidemiology, molecular testing, and therapeutic strategies for NSCLC with *HER2* molecular alterations.

## 1. Introduction

Over the past decade, genomic profiling has enabled the identification of numerous targetable oncogenic drivers involved in the tumor initiation and progression of multiple types of cancer [1,2]. More specifically, the therapeutic landscape of NSCLC has changed dramatically with the introduction of targeted agents, leading to unprecedented results in outcomes for patients [3].

The discovery of mutations in the epidermal growth factor receptor (EGFR) in NSCLC, as well as recurrent mutations in B-RAF proto-oncogene serine/threonine kinase (BRAF) and rearrangements in anaplastic lymphoma kinase (ALK)/ROS proto-oncogene 1 receptor tyrosine kinase (ROS1), has led to the development of targeted therapies with a marked impact on both the prognosis and the quality of life of lung cancer patients [4,5,6,7,8]. More recently, other agents targeting genomic alterations in RET, MET, and NTRK have given promising results and have led to the investigation of their status based on recent international guidelines [9].

These therapies are now the standard of care in oncogene-driven NSCLC and have stimulated the study of other mutations in kinases involved in NSCLC [10,11]. Among them, HER2/neu (encoded by the *HER2* gene) has been extensively studied in solid tumors. HER2 deregulation can occur through multiple mechanisms including gene amplification, mutations, and protein overexpression in a large number of breast and gastric adenocarcinomas [12]. Activating mutations in *HER2* are rare in NSCLC and occur in less than 5% of cases, most commonly in the adenocarcinoma subtype and in patients with no smoking history [13,14,15]. Based on the successful history of anti-HER2 therapy with trastuzumab in HER2-overexpressing and/or amplified breast cancer, several authors focused their interest on *HER2* aberrations in advanced NSCLC (aNSCLC) [16,17]. However, the evaluation of potential clinical implications of anti-HER2 agents has so far led to conflicting results [18,19,20]. Nevertheless, recent trials evaluating the antibody-drug conjugates (ADCs) ado-trastuzumab emtansine and, notably, trastuzumab deruxtecan in 2022 in patients with *HER2*-mutant aNSCLC have provided promising data [21,22,23]. So, these new agents are bringing new hope to the management of *HER2*-altered aNSCLC.

Subsequently, discussions about the combinations of agents with distinct mechanisms of action (i.e., irreversible tyrosine kinase inhibitors (TKIs) or immune checkpoint inhibitors (ICIs)) have rapidly taken place to improve the therapeutic options of HER2-driven NSCLC. In addition, new challenges are emerging concerning the detection of *HER2* alterations in these settings [24].

In this review, we describe and discuss *HER2* alterations in NSCLC including the diagnostic challenges and therapeutic options.

## 2. Biology of the HER2 Receptor

The HER2 (also known as Neu) protein is a 185 kDa transmembrane glycoprotein encoded by the *HER2* (*ERBB2*) gene, which is located on chromosome 17q12. It belongs to the EGFR family of receptor tyrosine kinases that includes HER1 (EGFR/ERBB1), HER2 (ERBB2), HER3 (ERBB3), and HER4 (ERBB4), which are activated by ligand-induced dimerization [25,26]. Together these receptors regulate key cellular processes including proliferation, motility, and survival. The HER2 receptor comprises three parts: the extracellular region containing four domains; the transmembrane part; and the intracellular tyrosine kinase domain (TKD) [25]. Signaling through the HER receptors involves a succession of steps leading to their activation: binding of the ligand, a conformational change in the receptor resulting in homo- or hetero-dimerization, transphosphorylation of the TKD of both partners of the dimerized receptor, and subsequent activation of downstream signaling cascades [27]. In contrast to other members of the EGFR family, HER2 has no known ligand, making it the preferred hetero-dimerization partner of all ERBB proteins [28].

After binding to an extracellular ligand, these receptors regulate cell proliferation and survival through three major signaling pathways: the Ras/Raf/MAPK and PI3K/Akt/mTOR pathways and the Janus kinase/signal transducer and activator of transcription (JAKSTAT) pathway through tyrosine kinase activation [29,30]. It is also a proto-oncogene that can modulate the process of carcinogenesis [12].

## 3. *HER2* Alterations and Carcinogenesis

The role of *HER2* in human carcinogenesis was first established when it was reported that about 20–30% of breast carcinomas display *HER2* amplifications and HER2 protein overexpression, which was associated with aggressive biological behavior and poor patient outcomes [31]. Consequently, the development of specific anti-HER2 therapies has had a huge impact on the management of patients with breast cancer. More recently, *HER2* amplification/overexpression has also been identified in other solid tumors, such as gastric carcinomas (showing up to 20% with *HER2* alterations), as well as in colorectal, salivary gland cancers, or uterine serous carcinomas, where anti-HER2 targeted therapy has also become the standard of care [32,33,34,35].

*HER2* amplification/overexpression is the most frequent *HER2* alteration in cancers [14]. It has been reported that HER2 induces cell proliferation and the invasion of non-small-cell lung cancer by upregulating COX-2 expression via the MEK/ERK signaling pathway. In addition, the expression of HER2 can promote the development of advanced and metastatic non-small-cell lung cancer [36,37]. However, advances in molecular biology and sequencing techniques have more recently identified a second mechanism concerning the major oncogenic activation of *HER2*: the occurrence of activating somatic mutations that results in tumor transformation. These activating mutations concern about 20% of all the *HER2* alterations observed in solid tumors [38].

Most of these mutations are in the extracellular domain (ECD) and TKD (about 95% of them), whereas mutations in the transmembrane domain (TMD) and juxtamembrane domain (JMD) together account for less than 15% of all *HER2* mutations in cancer [39]. Studies have shown that the main *HER2* mutations occur in the TKD (about 46%), affecting exon 20 (20%), exon 19 (11%), and exon 21 (9%). The most common mutations include Y772dupYVMA (6% of all *HER2* mutations) and L755P/S (5% of all *HER2* mutations) in the TKD. Mutations in the ECD occur in 37% of all cancers, where S310F/Y is the most frequent (11% of all *HER2* mutations) [12,40,41]. Missense mutations and in-frame insertions within the TKD often lead to increased kinase activities, whereas mutations in the TMD bring about increased protein stabilization [42]. The rate of *HER2* alterations and mutations has been shown to vary greatly across cancer types [41].

## 4. Epidemiology of *HER2* Alterations in NSCLC

### 4.1. HER2 Amplification/Overexpression in Lung Cancer (HER2-Positive NSCLC)

Unlike in breast and gastric carcinomas, the definition of HER2 positivity, *HER2* amplification, or HER2 overexpression in lung cancer remains unclear. Studies generally use a *HER2*/CEP ratio > 2 and/or a *HER2* copy number ≥ 6.0 to define *HER2* amplification with a reported wide range of prevalence from 2 to 20% [43,44]. For HER2 protein overexpression, an H-score (with a threshold of ≥200) or a semi-quantitative assessment according to the breast method are used preferentially (0, 1+, 2+, or 3+ based on membranous staining) [45,46]. Due to the lack of a method of consensus, the prevalence of HER2 overexpression varies considerably in lung cancer patients, ranging from 6 to 30% [43,47]. The prognostic impact of HER2-positive NSCLC is still equivocal, although HER2 overexpression/amplification seems to be associated with reduced disease-free survival and poor outcomes, especially for adenocarcinomas, early-stage NSCLC, and small-cell lung cancer (SCLC) [45]. *HER2* amplification and HER2 overexpression are detected more frequently in smokers and male patients, suggesting different origins of oncogenesis with *HER2* mutations [48]. *HER2* amplification has been identified as an acquired mechanism of resistance to osimertinib, mostly in the second-line setting [49]. Kaplan–Meier plots of the MSKCC 2020 lung adenocarcinoma cohort (available dataset on cBioportal.org) regarding the survival curves of patients with an *EGFR* mutation, *HER2* mutation, or *HER2* amplification are shown in Figure 1.

### 4.2. HER2 Mutations in Lung Cancer (HER2-Mutant NSCLC)

In NSCLC, *HER2* mutations mainly occur in exon 20 (about 50%), where the Y772dupYVMA mutation is the most frequent of the *HER2* mutations. Conversely, the mutation in the leucine residue at position 755 in exon 19 (L755) is the most common mutation in breast cancer, and the V842I variant in exon 21, as well as mutations in the ECD, are the most frequent in colorectal cancer [41,50].

The proportion of *HER2* mutations in lung cancer varies among the studies, with reported rates of 1–7%, and usually 2 to 4% [51,52,53]. In addition, the frequency in the type of *HER2* mutation shows large variations across the studies, and populations with a higher rate were observed in Asians (up to 7%) [45,54]. As mentioned above, an exon 20 insertion within the TKD is the most common *HER2* mutation in NSCLC. Similar to *EGFR*, insertions are in-frame, ranging from 3 to 12 base pairs, and are grouped in the proximal region of the exon. *HER2* insertions are less heterogeneous than their *EGFR* counterparts, with most of the mutations (34–83% depending on studies) leading to an insertion of 12 base pairs and the duplication of amino acids YVMA at codon 775, therefore named *HER2*^YVMA^ (A775_G776insYVMA insertion/duplication). After *HER2*^YVMA^, G778_P780dup and G776delinsVC mutations in exon 20 are the most frequent alterations [41,55,56,57].

*HER2* exon 20 and exon 19 in NSCLC also include uncommon missense mutations, notably in the V777, L755, G776, and D769 positions (variable incidence, up to 8–10% of all *HER2* mutations in cancer) [39,58].

Rarer mutations along the protein domains outside the TKD have also been reported, where S310F in the extra-cellular domain is the most frequent (about 5–7% in NSCLC) [59,60]. In addition, rare mutations affecting the TM and the JM domains (I655V, G660D, R678Q, E693K, and Q709L) have been recently reported [41,61].

Cell cycle alterations, particularly *TP53* abnormalities, are the most prevalent co-mutations in *HER2*-mutated NSCLC patients, followed by alterations in the PI3K pathway. Both were associated with shorter progression-free survival on afatinib treatment [56,62].

Regarding the clinicopathological aspects, *HER2* mutations in lung cancer are significantly associated with female sex, never smoking, and an adenocarcinoma histological subtype, such as that observed in *EGFR*-mutated NSCLC patients. However, *HER2*-mutated patients seem to have a poorer prognosis than their *EGFR* and *ALK* counterparts, with *HER2*^YVMA^ patients having the worst outcomes [43,48,60]. Studies have reported a variable prevalence of brain metastasis at diagnosis of *HER2*-mutant NSCLC (from 9 to 30%), which seems to be less frequent in *HER2*-mutant compared to WT *HER2* patients [43,57,63,64]. However, patients exhibited a trend toward brain metastases on treatment, with 28% of *HER2*-mutated NSCLC having brain involvement *versus* 8% for *KRAS*-mutated NSCLC [63]. Similarly, the *HER2*Y^VMA^ subtype has been associated with a higher estimated 12-month brain metastasis incidence compared with the non-YVMA group (40.2% vs. 3.6%) [65]. In addition, it was reported that *HER2* mutations were acquired in 1% of first-line osimertinib-treated patients and could represent a mechanism of resistance [66].

In contrast to breast cancer and although few patients harbor both a *HER2* amplification and mutation, *HER2* mutations in most cases are not associated with amplification or overexpression, suggesting a distinct entity and, thus, different therapeutic targets [46]. In fact, overexpression of the *HER2* protein occurs infrequently in NSCLC and is often due more to polysomy than to amplification [67,68]. The distributions of the *HER2* mutations in NSCLC are shown in Figure 2.

*HER2* mutations and other oncogenic drivers, such as *EGFR*, *KRAS*, *NRAS*, *ALK*, *PI3KCA,* and *BRAF*, have been shown to be mutually exclusive [57,69].

**Figure 2 jpm-12-01651-f002:**
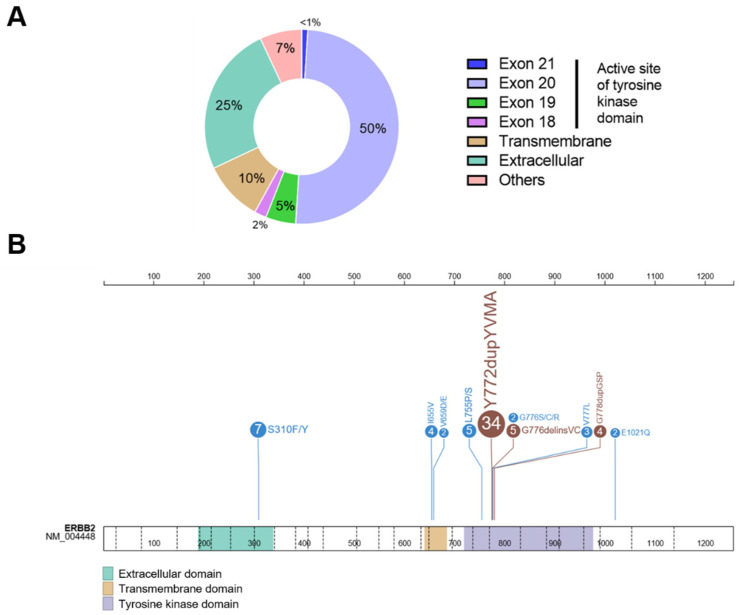
Distribution of the most frequent *HER2* mutations in NSCLC. (**A**). Plot displaying the protein domain and exon location of *HER2* mutations in NSCLC. (**B**). Lolliplot representation of *HER2* protein with frequency and location of the main *HER2* mutations in NSCLC. Numbers in circles are %. Brown circles are in-frame mutations. Blue circles are missense mutations. The data on frequency are from Robichaux et al., Fang et al., Zhou et al., and Li et al. [22,41,56,70].

## 5. Detection Methods for *HER2* Alterations in NSCLC

### 5.1. HER2 Amplification/Overexpression Detection

The distinction between *HER2* amplification and overexpression remains vague because of the number of testing methods and the different definitions of *HER2* positivity [71].

In routine clinical practice, next-generation sequencing (NGS) is commonly used for the detection of *HER2* amplification in NSCLC. However, there is no uniform standard for defining amplification across NGS platforms. Nevertheless, fluorescence in situ hybridization (FISH) remains the gold-standard method for *HER2* gene amplification detection and is recommended in clinical studies [44]. For HER2 overexpression testing, immunohistochemistry (IHC) is recommended as a standard method for solid tumors including NSCLC [44] (Table 1).

The most accepted definition of *HER2* amplification is an average ratio of the *HER2* gene copy number to centromeres [*HER2*/chromosome enumeration probe 17 (CEP17)], that is, 2 and/or a *HER2* copy number ≥ 6.0 by FISH [43,44]. In addition, the IHC scoring system ranging between 0 and 3+, which is based on membrane staining (with IHC 0–1+ defined as a HER2-negative and IHC 2+–3+ as a HER2-positive tumor), remains the most frequently used method to detect HER2 overexpression in lung cancer [44,45]. An H-score (with a threshold ≥ 200) can also be used. However, HER2 protein expression cannot be used as a surrogate marker for *HER2* mutations or amplifications in lung cancer [46].

### 5.2. HER2 Mutation Detection

The methods used to evaluate *HER2* mutations mainly include Sanger sequencing, NGS, and droplet digital PCR (ddPCR) [44] (Table 1). These methods have different sample requirements and vary in the types of genetic alterations tested, difficulty of operation, and turnaround time (TAT). The selection of the testing method must be made based on the organization of the local laboratory, the sample type and size, as well as the clinical needs. NGS requires less DNA for testing than other methods and so is optimal for small thoracic specimens (small biopsy, fine-needle biopsy, and cytological samples) or liquid biopsies [24,72,73]. Thus, an NGS platform should be able to identify all types of variations in *HER2* including all exon 20 insertions; missense mutations in TKD, JMD, and ECD; copy number variations and amplifications with a low requirement of input DNA; high repeatability; and low TAT. The latest ESMO guidelines recommend NGS for *HER2* mutation testing and strongly suggest testing at least for *HER2* exon 20 mutations in unresectable stage III and stage IV NSCLC that meet two or three of the following criteria: (a) adenocarcinoma or adenosquamous carcinoma of the lung; (b) no or mild smoking history; and (c) female patient [44] (Table 1).

**Table 1 jpm-12-01651-t001:** Techniques for detecting *HER2* molecular alterations and recommendations for interpretation.

	Main Method (Interpretation)	Alternative Methods (Interpretation)
***HER2* mutation**	Sequencing techniques - Next-generation sequencing - Sanger - Pyrosequencing	RT-PCR qPCR
***HER2* amplification**	FISH (*HER2*/CEP17 ratio >2 and/or *HER2* copy number > 6)	NGS (gene copy number > 6)
**HER2 overexpression**	IHC based on membranous staining according to breast guidelines [74]: **>0:** *HER2* negative; **>1+:** needs to be confirmed by further studies whether 1+ should be considered negative or as having a *HER2* low expression; **>2+, 3+:** *HER2* positive. Due to the poor concordance between FISH and IHC in NSCLC, FISH confirmation is not required for NSCLC patients with IHC 2+/3+ to define positive *HER2* expression.	

IHC, immunohistochemistry; FISH, Fluorescent in situ hybridization.

## 6. HER2-Targeted Therapy in NSCLC

In recent years, studies, unfortunately, have failed to establish an association between *HER2* status and an objective response to chemotherapy regimens [47,75,76]. In addition, pemetrexed-based chemotherapy has even shown poorer outcomes in patients with *HER2*-mutant aNSCLC compared to patients with *ALK*/*ROS1* rearrangements [77]. Taken together, these data have encouraged the development of more effective HER2-targeted therapies. Recently, many studies have, therefore, been conducted to evaluate therapies specifically targeting HER2 (including anti-HER2 mAbs trastuzumab and pertuzumab alone or combined with chemotherapy) [64,78]. However, unlike breast and gastric cancers, these treatments have shown modest and inconstant results and are still not considered the standard of care for NSCLC. Fortunately, encouraging efforts in the development of novel treatments have been made and have given very promising results.

We describe here the emerging and now available anti-HER2 drugs while focusing on the most promising therapies according to the mechanism of action (i.e., selective TKI, and Antibody–Drug Conjugates (ADC), with a note regarding the use of ICIs.

### 6.1. Non-Selective HER2 Tyrosine Kinase Inhibitors

Second-generation irreversible TKIs were initially developed for patients with *EGFR*-mutated tumors and represented the first attempts to target HER2 in lung cancer. The structural similarities between the EGFR and HER2 proteins explain why these drugs were subsequently investigated in *HER2*-mutant NSCLC despite their lack of specificity for HER2. Some retrospective studies have raised hope for the use of chemotherapy and TKI in *HER2*-mutant NSCLC [79,80]. However, non-selective HER2 TKI targeting both EGFR/HER2 such as afatinib and pan-HER TKI such as dacotinib and neratinib revealed a low efficacy and did not reach the objectives needed for broad use as the standard of care in prospective trials. Indeed, the objective response rate (ORR) for these TKIs ranged from 0 to 19%, mainly in small phase II studies (cohorts of 7 to 60 patients with most studies involving less than 30 patients) [51,81,82,83,84].

### 6.2. Selective HER2 Tyrosine Kinase Inhibitors

Recently, more selective pan-HER TKIs have been designed especially for *EGFR*-mutant NSCLC. Compared with non-selective TKIs, these novel TKIs have shown greater and broader anti-tumor effects by binding particularly well to exon 20 mutations in the HER protein family including HER2. These treatments have raised new hope for patients with advanced cancers who had previously received platinum-based chemotherapy at a time when therapeutic options were limited.

#### 6.2.1. Poziotinib

Poziotinib is a covalent and irreversible EGFR/HER2 inhibitor with a smaller size and flexible structure compared to afatinib and allows circumventing the hindered binding pocket of exon 20 insertions [85]. In a preclinical study comparing the activity of different TKIs in vitro and in patient-derived xenograft (PDX) models with *HER2* exon 20-mutant NSCLC, poziotinib was more effective than other HER2 TKIs. In addition, an in vitro study suggested that the secondary C805S mutation could be a potential mechanism of acquired resistance to poziotinib [85,86].

In 2018, the first in-human phase I trial that examined the safety of poziotinib in 75 patients with advanced solid tumors including aNSCLC showed a tolerable toxicity level, therefore supporting further clinical development of poziotinib and its application in aNSCLC with *HER2* mutations [87].

Based on this study, a phase II clinical trial evaluating poziotinib in aNSCLC patients with *EGFR* and *HER2* exon 20 mutations was initiated. Twelve *HER2*-mutant participants were enrolled and all had Y772dupYVMA or G778dupGSP insertions. The early results demonstrated an ORR of 42% (5/12 patients) with a duration of response exceeding 1 year and a median Progression-Free Survival (PFS) of 6 months. Moreover, the treatment was well tolerated with no dose reduction due to drug-related toxicity [41]. Additionally, a single-arm, open-label, phase II study was conducted to assess the efficacy and safety profiles of poziotinib in *HER2*-mutant advanced NSCLC. The ORR, median PFS, and Overall Survival (OS) were 27%, 5.5, and 15.0 months, respectively. However, one possible treatment-related death due to pneumonitis was reported [88]. Finally, an expanded access program evaluating poziotinib in aNSCLC patients with an *EGFR*/*HER2* exon 20 insertion showed a median PFS of 5.6 months and a median OS of 9.5 months. The ORR was higher in the *HER2* subgroup (50% vs. 23%). However, frequent grade 3 Adverse Events (AE) (66% of the patients) led to a high rate of dose interruption and reduction [89].

More recently, the multinational, multicohort phase II ZENITH20 study assessed 16 mg of poziotinib once a day in previously treated and treatment-naïve NSCLC patients with *HER2* exon 20 insertions. In cohort 2, 90 patients were enrolled, 98% of whom had prior chemo/platinum-based therapy; 67% had immunotherapy including ICIs; and 28% had HER2 therapy. The ORR and Disease Control Rate (DCR) were 27.8 and 70.0%, respectively. A total of 74% of patients had tumor reductions, with a median PFS of 5.5 months. Greater responses (39%) were observed in patients who were heavily pre-treated (≥3 prior treatment lines), with no association between *HER2* mutation variants and clinical outcomes. Severe treatment-related AEs (grade ≥ 3) were frequently observed and included rash (49%), diarrhea (27%), and stomatitis (24%) [90]. In cohort 4, 56 treatment-naïve patients with *HER2* exon 20-insertion aNSCLC were enrolled to receive 8 mg poziotinib twice a day. The ORR and median PFS were 44% and 5.6 months, respectively, with reduced toxicity compared to the 16 mg cohort [91]. Accordingly, the FDA is currently discussing potential approval.

#### 6.2.2. Pyrotinib

Pyrotinib is a small-sized covalent pan-HER inhibitor derived from 3-cyanoquinoline. Its configuration makes it superior to afatinib regarding efficacy and selectivity in in vitro and in vivo models (including *HER2*-mutant NSCLC patient-derived organoids and PDX murine models) [92]. In the same study, the results from a phase II cohort of 15 *HER2*-mutant NSCLC patients treated with pyrotinib showed an ORR and a median PFS of 53.3% and 6.4 months. No grade 3 or higher AEs were observed with no occurrences of dose reductions. A large phase II trial including 60 patients with refractory NSCLC with *HER2* mutations revealed an ORR of 30%, with a median Duration of Response (DoR) of 6.9 months and a median PFS and OS of 6.9 and 14.4 months, respectively [70]. All subgroups of patients with different types of *HER2* mutations showed a favorable objective response rate. However, the ORRs were significantly higher in patients with a 12-bp or 9-bp exon 20 insertion, than in patients with G776 or L755P mutations (27 and 60% vs. 17 and 25%, respectively). No ORR difference was observed in the cases of brain metastasis (25 vs. 31.3%). Grade 3–4 AEs occurred in 28% of patients and diarrhea was the most frequent type.

Interestingly, data from another single-arm prospective study evaluating pyrotinib in 27 advanced *HER2*-amplified NSCLC patients revealed good efficacy and safety with an ORR of 22.2%, a median PFS of 6.3 months, and a median OS of 12.5 months. In the subgroup of patients who received pyrotinib as first-line treatment, the median PFS was 12.4 months. Moreover, 30.8% of the patients who had progressed after EGFR TKI treatment responded to pyrotinib [93].

Given all these encouraging results, a phase III study evaluating pyrotinib versus docetaxel as second-line therapy in patients with advanced non-squamous NSCLC harboring a *HER2* exon 20 mutation (PYRAMID-1/ NCT04447118) is now recruiting patients.

#### 6.2.3. Tarloxotinib

Tarloxotinib is a hypoxia-activated prodrug of a pan-HER kinase inhibitor. It has been shown that tarloxotinib is converted into its active form tarloxotinib-E (a potent irreversible metabolite) in a hypoxic tumor microenvironment and it is also an NRG1 fusion inhibitor activating *HER2*/3. Preclinical studies have shown tarloxotinib-induced tumor regression in murine xenograft models with *EGFR* and *HER2*-mutant NSCLC. Tarloxotinib-E may interfere with cell signaling and proliferation by inhibiting the activation of HER2 heterodimers in PDX models. Subsequently, a major clinical response to tarloxotinib was observed in one patient with an *EGFR* exon 20 A775_G776insYVMA-insertion NSCLC [94].

Recently, the first phase I/II multicohort RAIN-701 trial (NCT03805841) evaluating the efficacy and tolerance of tarloxotinib among solid tumors enrolled 11 patients with *HER2*-mutant advanced chemotherapy pre-treated aNSCLC (cohort B). Among them, nine were evaluable. The initial results showed that a partial response (PR) was achieved for 22% of patients and 50% of patients had stable disease (SD). Most AEs were of grades 1 or 2. The most reported grade 3 AEs were prolonged QTc (35%) and increased alanine aminotransferase (ALT) (4%), leading to dose reduction and discontinued treatment in 22 and 4% of patients, respectively [95].

#### 6.2.4. Mobocertinib

Mobocertinib (TAK-788/AP3278) is an oral next-generation EGFR/HER2 inhibitor designed to target exon 20 insertions that irreversibly binds to EGFR via a covalent modification of the Cys797 residue in the active site of EGFR. In contrast to first- and second-generation TKI, mobocertinib displays a better inhibition and selectivity for all mutant variants of both *EGFR* and *HER2* than for wild-type *EGFR* (IC50 2.4–22 nM and 2.4–26 nM vs. IC50 35 nM, respectively). In addition, lung cancers of *HER2* G776delinsVC subtypes showed a superior response to mobocertinib than the YVMA subtypes [96].

In September 2021, the FDA accelerated approval of mobocertinib for metastatic NSCLC with *EGFR* exon 20 insertion mutations following the initial results of phase I/II study 101, an international, non-randomized, open-label, multicohort clinical trial (NCT02716116). This study recruited refractory NSCLC patients with *EGFR*/*HER2* exon 20 insertions. Although data for the efficacy of treatment of the *HER2*-mutant subtype are still being processed, the *EGFR*-mutant expansion cohort revealed encouraging data with PRs achieved in 12 out of 28 assessable patients (ORR 43%), giving a DCR of 86% and a median PFS of 7.3 months [97]. The safety profile was similar to that of other EGFR-TKIs. Grade 3–4 AEs were reported in 5% of patients, and these were mainly diarrhea.

### 6.3. Antibody–Drug Conjugates (ADC) against HER2

ADC are characterized by the covalent coupling of a cytotoxic payload to a monoclonal antibody (mAbs) that is directed toward a target antigen expressed on the cancer cell surface, reducing systemic exposure and therefore toxicity. Nowadays, the development of ADCs represents a breakthrough in the treatment of cancer with actionable alterations. They were initially developed for HER2-overexpressing tumors as the Abs are not selective for mutant HER2 but HER2 in general and thus should yield better efficacy in overexpressing tumors. However, the major success in breast and gastric cancers of ADCs has drawn attention to NSCLC with *HER2* alterations, including overexpression and, notably, mutations.

#### 6.3.1. Trastuzumab Emtansine

Trastuzumab emtansine (T-DM1) is a second-generation anti-HER2 ADC composed of trastuzumab and emtansine and the cytotoxic microtubule agent emtansine (DM1), a maytansine derivative. This complex penetrates HER2-positive cells via receptor-mediated endocytosis leading to the proteolytic degradation of the antibody moiety in lysosomes and the release of conjugated agents [98].

In a phase II study of T-DM1 carried out with 15 relapsed HER2-positive NSCLC (IHC 3+, IHC 2+/FISH+, or exon 20 mutations), only one *HER2*-mutant patient achieved a PR (global ORR = 6.7%). The median PFS and OS were 2.0 and 10.9 months, respectively. No responses were obtained in the *HER2*-amplified/overexpressing subgroup. Frequent grade 3–4 AEs were associated (including thrombocytopenia (40%) and hepatotoxicity (20%)); however, no dose reductions or treatment discontinuations occurred. The limited efficacy resulted in the early termination of the study [21].

Subsequent results from another T-DM1 phase II clinical trial showed that 8 out of 18 patients with advanced HER-mutant NSCLC achieved PRs. The ORR was 44% with a median PFS of 5 months. A response to T-DM1 was observed across all the subtypes. The toxicity included grade 1–2 elevated hepatic transaminases, thrombocytopenia, and infusion reactions [99]. Updated data with 28 pre-treated patients showed an ORR of 50% [100]. A cohort of 11 patients with *HER2*-amplified NSCLC included in the latter basket trial reached an ORR of 55% [100]. A subsequent study focusing on 49 HER2- overexpressing NSCLC (29 IHC 2+ and 20 IHC 3+) found an ORR of 20% in the IHC3+ subgroup compared to no response in the IHC2+ subgroup, whereas the median PFS and OS were similar (2.6 vs. 2.7 months and 12.2 vs. 5.3 months, respectively). Ten patients (20%) reported grade 3 AEs but no deaths due to AEs occurred [101]. Finally, a very recent study that included 22 *HER2*-mutant NSCLC showed an ORR of 38.1% and a DCR of 52.4%. The median PFS and median OS were 2.8 and 8.1 months, respectively, with mild toxicity [102].

#### 6.3.2. Trastuzumab Deruxtecan

Trastuzumab deruxtecan (T-Dxd/DS-8201a) is a novel HER2-ADC composed of trastuzumab and a topoisomerase I inhibitor (MAAA-1181) linked to an enzymatically cleavable peptide with a different mechanism of action to other ADCs. The drug moiety of T-Dxd binds to and stabilizes topoisomerase I-DNA complexes inducing DNA double-strand breaks. Its drug-to-antibody ratio (DAR) is 8, which is twofold higher than T-DM1 (DAR of 3–4), allowing the steady delivery of the topoisomerase I inhibitor, even in the case of *HER2* low expression. Featured by a highly membrane-permeable payload, it is favorable in treating tumors that are insensitive to T-DM1 or *HER2*-negative tumors [103,104].

The early data on T-Dxd in NSCLC originated from a solid non-breast and non-gastric cancers dose-expansion phase I study that enrolled 18 patients with *HER2*-mutant/*HER2*-expressing NSCLC. T-Dxd had great potential with 10 of 18 patients (55.6%) experiencing a confirmed objective response and a median PFS of 11.3 months. Among the subset of *HER2*-mutant NSCLC patients, the ORR reached 72.7% (8/11) and the median PFS was 11.3 months. Two out of eighteen patients (11.1%) had serious AEs and three patients were diagnosed with interstitial lung disease (ILD) related to the drug. One of them had a fatal outcome due to this respiratory AE [105].

Very recently, the most promising and exciting results came from the final analysis of the multicenter phase II trial, DESTINY-Lung01, which evaluated the efficacy of T-Dxd in refractory NSCLC with *HER2* molecular alterations. Cohort 1 enrolled 49 patients with HER2 overexpression (IHC 2+ or IHC 3+), and cohort 2 enrolled 91 patients with *HER2* mutations. In cohort 1, the ORR and median PFS were 24.5% and 5.4 months, respectively. The response rates were comparable according to the HER2 IHC expression levels. Grade 3 AEs were reported in 73.5% of patients. As previously reported, drug-related ILD was adjudicated for eight cases of drug-related ILD. AEs were associated with dose interruptions in 53.1% of patients, dose reductions in 34.7%, and treatment discontinuations in 22.4% [106]. However, the most encouraging results came from cohort 2. Indeed, the confirmed ORR was 55%, the median PFS was 8.2 months, and the median OS was 17.8 months [22]. In this cohort, 85% (78/91) of patients had a *HER2* exon 20 insertion, 8% (7/91) had a punctual mutation in the *HER2* TKD (exon 19 or 20), and 7% (6/91) had a punctual mutation in the *HER2* exon 8 ECD. Interestingly, patients with a *HER2* mutation in the exon 8 ECD seemed to respond less to T-Dxd, as the only three patients that progressed under therapy harbored this mutation.

*HER2*-activating mutations were found to facilitate the endocytosis of the HER2-ADC complex providing a potential explanation for the higher efficacy in *HER2*-mutant NSCLC patients in contrast to *HER2*-overexpressing patients [107]. Grade 3 or higher AEs were observed in 42 patients (46%) and 23 patients discontinued treatment. It is worth noting that ILD occurred in five patients with no drug-related deaths. These very encouraging results from DESTINY-Lung01 have rapidly motivated the setup of a randomized, open-label, phase III trial (DESTINY-Lung04; NCT05048797) to evaluate the efficacy and safety of T-Dxd compared to the standard of care (pembrolizumab combined with chemotherapy) in patients with non-squamous NSCLC harboring a *HER2* exon 19 or 20 mutation. Very recent preliminary data from DESTINY-Lung02/NCT04644237 (a phase II randomized dose-finding trial) showed that the ORR among a cohort of *HER2*-mutant NSCLC patients receiving 5.4 mg/kg of T-Dxd (*n* = 52) was 57.7%. Complete responses were observed in 1.9% of patients and partial responses were observed in 55.8%. The median duration of responses was 8.7 months. Following all these results, T-Dxd was granted accelerated approval in August 2022 by the FDA for adult patients with previously treated, unresectable, or metastatic NSCLC, whose tumors have an activating *HER2* mutation. Thus, T-Dxd is the first drug approved for *HER2*-mutant NSCLC, which represents an important milestone for patients and the healthcare community. Further analyses from the DESTINY-Lung02 trial will be discussed at upcoming medical meetings.

### 6.4. Immune Checkpoint Inhibitors (ICI)

Limited data concerning ICIs as a single-agent therapy has come from retrospective studies reporting that the ORR for patients with advanced *HER2*-mutant lung cancer ranges from 7.4 to 29%, with a PFS ranging from 1.8 to 3.4 months [108,109,110,111,112].

In the IMMUNOTARGET registry, 29 *HER2*-mutant NSCLC patients treated with single-agent ICIs had a PFS of 2.5 months and an OS of 20.3 months. None of them had ≥50% programmed cell death ligand-1 (PD-L1) staining [112]. Another study from the French Lung Cancer Group (GFPC) revealed that 6/23 patients with *HER2*-mutant relapsed NSCLC responded to ICIs (ORR = 27.3%), with a median PFS OS and DoR of 2.2, 20.4, and 15.2 months, respectively. However, the PD-L1 status was unknown for 65% of patients [111].

In 2021, a study compared the influence of the immune microenvironment of patients with *HER2* mutations with those with *EGFR* mutations. Patients with *EGFR* exon 20 insertions had a significantly higher level of PD-L1 expression than those with *HER2* mutations, which might account for their improved response to immunotherapy [113].

In addition, regimens of combined ICIs and chemotherapy have been explored and evaluated. A multicenter retrospective study showed an ORR, DCR, and median PFS of 38.5%, 84.6%, and 7.4 months, respectively for 26 patients with *HER2*-mutant NSCLC, including 16 patients treated with immunochemotherapy combination regimens [114]. In addition, another study into treatment-naive patients receiving ICIs in combination with chemotherapy found that the ORR, median PFS, and OS rate at one year were 52%, 6 months, and 88%

The results from another study that treated 27 patients with first-line ICIs in combination with immunochemotherapy demonstrated an ORR, a median PFS, and a one-year OS rate of 52%, 6 months, and 88%, respectively [115]. Finally, another retrospective study highlighted that patients with a high baseline tumor mutational burden (TMB) and mutations in DNA damage repair-related pathways or the SWI/SNF complex seem to be associated with favorable outcomes of chemo-immunotherapy combinations [116].

Taken together, none of these retrospective studies seem to favor the use of ICIs as monotherapy for patients with *HER2*-mutant NSCLC. Thus, consistent data from large prospective trials are needed, especially for immuno-chemotherapy combination regimens, that may produce more favorable results.

Clinical trials and the efficacy of anti-HER2 agents and ICIs in patients with NSCLC with *HER2* alterations are displayed in Table 2.

## 7. Perspectives and Conclusions

Considerable progress has been made in the management of NSCLC with the identification of novel therapeutic biomarkers and targets that continue to emerge.

Among them, *HER2* alterations come across as promising actionable targets that provide interesting, but limited, data from previous and ongoing clinical trials. The treatment landscape of this very heterogeneous subset of NSCLC is rapidly changing as a result of the discovery of new drugs currently under clinical evaluation. However, a number of concerns need to be discussed.

First, NSCLC with *HER2* alterations display a biological and clinical heterogeneity that may be responsible for the limited and variable efficacy of HER2-targeted therapies. Most of the studies concern small cohorts of patients and evaluate anti-HER2 therapies in *HER2*-mutant and HER2-positive NSCLC (amplification and/or overexpression) without distinction, which impairs the interpretation of their predictive values. Given that there is no clear association between *HER2* mutations and amplification or overexpression, questions surrounding the definition of ‘HER2-positive’ NSCLC and the type of ‘HER2-positive’ aNSCLC eligible for targeted therapy remain to be addressed. This may explain the variation in the rate of IHC overexpression/amplification reported in the different studies and the limited efficacy of anti-HER2 drugs in this setting. Thus, *HER2* expression/copy number assessment cannot currently be a reliable biomarker of response to targeted therapies in NSCLC.

The type of *HER2* mutation and their respective responses to targeted therapies are rarely specifically evaluated in clinical trials. Some studies suggest a better response to anti-HER2 drugs targeting TKD mutations compared to other domain mutations, although the responses do not seem comparable between each mutation/exon within the TKD [22,70,96]. To define *HER2* alterations and determine their response patterns, it is mandatory to have worldwide standardized methods of detection (thresholds for HER2 overexpression/amplification, type of *HER2* mutation, and exons covered by NGS panels). Moreover, patient subgroups in further trials should be better defined according to their type of alteration and mutation to determine the population that may best benefit from each anti-HER2 therapy.

Brain metastases are common in *HER2*-mutant NSCLC at diagnosis and during treatment. Preliminary data regarding the activity of the central nervous system (CNS) with novel TKIs have shown promising results [70,91]. First-line pyrotinib has even been shown to induce a prolonged CNS response in a patient with *HER2*-mutant NSCLC [117]. Therefore, frequent monitoring for the early identification of brain metastasis along with the iterative follow-up of the CNS activity of anti-HER2 drugs is ongoing but future studies are needed.

Another promising perspective is the development of combined therapies that includes agents with synergistic mechanisms of action such as the combination of ADCs with irreversible TKIs or ICIs. Preclinical studies have provided evidence for the rationale behind combination therapy in patients with *HER2*-altered NSCLC. For instance, T-Dxd has been shown to increase tumor-infiltrating CD8+ T cells and expression of PD-L1 along with Major Histocompatibility Complex-Class I on tumor cells. These findings suggest that program cell death-1 (PD-1) inhibitors would be more effective in association with ADCs than alone. Thus, several phase I/II clinical trials are currently evaluating the tolerability and efficacy of such therapies (NCT03334617, NCT04686305, and NCT04042701). The association of ADCs with selective anti-HER2 TKI is also an interesting strategy. It has been reported in in vitro and in vivo studies that the combination of T-DM1 with a pan-HER inhibitor enhanced receptor ubiquitination and consequent internalization of HER2-ADC complexes, leading to a potent anti-tumor activity. In addition, the use of T-Dxd as second-line therapy could provide a durable response in the case of resistance to T-DM1 [85,107].

Finally, special attention should be paid to the molecular alterations associated with *HER2*. Some of them (i.e., *TP53*) are not unusual and can impair the efficacy of anti-HER2 agents resulting in a poorer prognosis. Moreover, *HER2* alterations are reported to drive EGFR-TKIs’ resistance. Concomitant treatment of EGFR-TKIs and anti-HER2-targeted therapies should then be considered. Further investigations are needed to better understand the molecular landscape of NSCLC with *HER2* alterations and its impact on drug resistance and patient outcomes.

In the last few decades, the treatment of lung cancer has improved substantially with the arrival of new therapies. Only a few years ago, NSCLC with *HER2* alterations were considered poor targets with few treatment options and a poorer outcome compared to those with other molecular alterations.

Nowadays, *HER2*-altered NSCLC are set for a revolution with plenty of pre-clinical and clinical findings exploring new treatments and strategies that are giving very promising results. Although the wide array of emerging therapies requires further clinical validation, the future seems brighter and full of hope for these patients.

Thus, all the novel data and breakthroughs support the need to develop standardized and systematic *HER2* alteration reflex testing for the diagnosis of NSCLC.

## Figures and Tables

**Figure 1 jpm-12-01651-f001:**
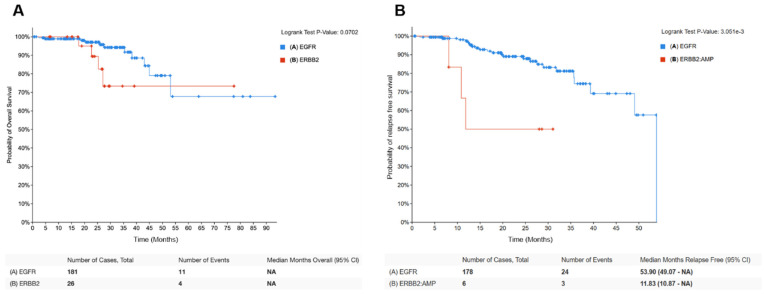
Kaplan–Meier curves and univariate survival analysis (log-rank test) of patients from the MSKCC 2020 lung adenocarcinoma cohort (604 patients). (**A**). Overall survival of patients with *EGFR*-mutant vs. *HER2*-mutant lung adenocarcinomas (**B**). Relapse-free survival of patients with *EGFR*-mutant vs. *HER2*-amplified lung adenocarcinomas.

**Table 2 jpm-12-01651-t002:** Main clinical trials and efficacy of treatment in NSCLC patients with *HER2* alterations.

Class	Drugs	Clinical Trial	Population	Cohort Size (*n*)	*HER2* Alteration	ORR *n* (%)	DCR *n* (%)	Median PFS, Months (95% CI)	Median OS, Months (95% CI)	References
**Selective TKI**	**Poziotinib**	Phase II study (NCT03066206)	Metastatic, recurrent NSCLC	12	*HER2* mutation (Y772dupYVMA (9) Or G778dupGSP (3))	5 (42)	10 (83)	5.6 (NA)	NA	Robichaux et al [41].
**Selective TKI**	**Poziotinib**	Phase II study (NCT03066206).	Stage IV or recurrent NSCLC, 90% of patients were pretreated	30	*HER2* mutation (Y772_A775dupYVMA (23), G778_P780dupGSP (5) or G776delinsVC (2))	8 (27)	22 (73)	5.5 (4.0–7.0)	15 (9.0–NE)	Elamin et al. [88].
**Selective TKI**	**Poziotinib**	Phase II study, expanded access program	Advanced NSCLC	8 ^a^	*HER2* exon 20 insertion	4 (50)	6 (75)	5.6 (3.6–6.7) ^b^	9.5 (5.3–NE) ^b^	Prelaj et al. [89]
**Selective TKI**	**Poziotinib**	Phase II Basket trial ZENITH20 study (NCT03318939)	Pretreated, advanced NSCLC	90 (cohort 2)	*HER2* mutation (Y772_A775dupYVMA (65), G776delinsVC (11), G778_P780dupGSP (7) or other mutant (7))	25 (27.8)	63 (70)	5.5 (3.9–5.8)	NA	Le et al. [90].
**Selective TKI**	**Poziotinib**	Phase II Basket trial ZENITH20 study (NCT03318939)	Treatment naïve, advanced NSCLC	48 (cohort 4)	*HER2* exon 20 insertion	21 (44)		5.6 (NA)	NA	Cornelissen et al. [91].
**Selective TKI**	**Pyrotinib**	Phase I/II study (NCT02535507)	Pretreated, advanced NSCLC	15	*HER2* exon 20 insertion (A775_ G776insYVMA (10))	8 (53.3)	11 (73.3)	6.4 (1.6–11.2)	12.9 (2.1–23.8)	Wang et al. [92].
**Selective TKI**	**Pyrotinib**	Phase II, single-arm study (NCT02834936)	Pretreated, advanced NSCLC	60	*HER2* mutations (12-bp exon 20 insertion (44) G776 mutation (6) G778_P780dupGSP (5), L755P (4), or V777L (1))	18 (30)	51 (85)	6.9 (5.5–8.3)	14.4 (12.3–21.3)	Zhou et al. [70].
**Selective TKI**	**Pyrotinib**	Prospective, single-arm study (ChiCTR1800020262)	Stage IIIB/IV NSCLC	27	*HER2* amplification	6 (22.2)	18 (81.5)	6.3 (3.0–9.6)	12.5 (8.2–16.8)	Song et al. [93].
**Selective TKI**	**Tarloxotinib**	Phase II Basket trial RAIN-701 study (NCT03805841)	Progressive disease after platinum-based CT	11 (cohort B), 9 were evaluable	*HER2* mutations (not specified)	2 (22)	6 (67)	NA	NA	Liu et al. [95].
**ADC**	**T-DM1**	Phase II, single-arm study	Pretreated, advanced NSCLC	7/15	*HER2* mutation (A775_ G776insYVMA (5))	1 (4.3)	5 (71.4)	2.0 (1.2–4)	10.9 (4.4–12)	Hotta et al. [21].
				8/15	*HER2* amplification/ overexpression (IHC3+ or IHC2+ confirmed by FISH)	0 (0)	3 (37.5)	NA	NA	
**ADC**	**T-DM1**	Phase II Basket trial (NCT02675829)	Advanced NSCLC, 83% pretreated with CT	28/49	*HER2* mutation (subtypes not specified)	14 (50)	NA	5 (3.5–5.9)	NA	Li et al. [99,100].
				11/49	*HER2* amplification	6 (55)	NA			
**ADC**	**T-DM1**	Phase II, single-arm study	Locally advanced or metastatic NSCLC, pretreated with ≥1 CT	29	HER2 overexpression IHC2+	0 (0)	8 (28)	2.6 (1.4–2.8)	12.2 (3.8–23.3)	Peters et al. [101].
**ADC**	**T-DM1**	Phase II, single-arm study	Stage III/IV NSCLC pretreated with CT or NSCLC with postoperative recurrence	22	*HER2* exon 20 mutation (A775_ G776insYVMA (19))	8 (38)	11 (52)	2.8	8.1	Iwama et al. [102]
				20	HER2 overexpression IHC3+	4 (20)	8 (40)	2.7 (1.4–8.3)	15.3 (4.1–NE)	
**ADC**	**T-Dxd**	Phase I study (NCT02564900)		11	*HER2* mutation (44.4% exon 20 insertions)	8 (72.7)	10 (90.9)	11.3 (8.1–14.3)	17.3 (17.3–NE)	Tsurutani et al. [105].
**ADC**	**T-Dxd**	Phase II study Two-cohort and two-arm DESTINYLung01 (NCT03505710)	Pretreated, metastatic NSCLC	49	HER2 overexpression (IHC2+/3+)	12 (24.5)	34 (69.4)	5.4 (2.8–7.0)	11.3 (7.8–NR)	Nakagawa et al. [106].
			Pretreated, unresectable, or metastatic NSCLC	91	*HER2* mutation (exon 20 insertion, (78), mutation in *HER2* TKD exon 19 or 20 (7), or mutation in *HER2* ECD exon 8 (6))	50 (55)	84 (92)	8.2 (6.0–11.9)	17.8 (13.8–22.1)	Li et al. [22].
**Single ICI**	**Anti-PD1 (Nivolumab 89.6%)**	Retrospective study—IMMUNOTARGET Registry ^c^	Pretreated, advanced NSCLC	29	*HER2* mutation (not specified)	2 (7.4)	9 (31)	2.5 (1.8–3.5)	20.3 (7.8–NR)	Mazieres et al. [112].
**Single ICI**	**Anti-PD1 (Nivolumab 83%)**	Retrospective study—French Lung Cancer Group (GFPC) ^d^	Pretreated, advanced NSCLC	23	*HER2* mutation (not specified)	6 (27.3)	11 (50)	2.2 (1.7–15.2)	20.4 (9.3–NR)	Guisier et al. [111].
**ICI + CT**	**Combined therapy (pembrolizumab 80%)**	Retrospective study	Treatment-naïve, advanced NSCLC	27 (21 patients were assessable)	*HER2* mutation (Exon 20 Insertion (16), TKD Mutation (1), ECD mutation (4))	11 (52)	NA	6 (6–14)	NA	Saalfeld et al. [115].

ADC: Antibody–drug conjugate; ORR, objective response rate; PFS, progression-free survival; OS, overall survival; CT, chemotherapy; IHC, immunohistochemistry; NA: not available. NE: not estimable. ^a^ This phase II study enrolled 30 patients, 22 with *EGFR* 20 exon mutations, and 8 with *HER2* mutations. ^b^ Evaluation based on the whole cohort (*n* = 30). ^c^ Multicenter study of 551 patients in 24 centers from 10 countries. Molecular alterations enrolled were *KRAS* (*n* = 271), *EGFR* (*n*= 125), *BRAF* (*n* = 43), *MET* (*n* = 36), *HER2* (*n* = 29), *ALK* (*n* ¼ 23), *RET* (*n* = 16), *ROS1* (*n* = 7), and multiple drivers (*n* = 1). ^d^ Multicenter French study of 107 patients in 21 centers. Molecular alterations enrolled were *BRAF* (*n* = 44), *MET* (*n* = 30), *HER2* (*n* = 23), and *RET* (*n* = 9).

## Data Availability

No new data were created or analyzed in this study. Data sharing is not applicable to this article.

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
