# Peer review of "Deciphering the Impact of HER2 Alterations on Non-Small-Cell Lung Cancer: From Biological Mechanisms to Therapeutic Approaches"

_jpm, 2022, doi:10.3390/jpm12101651_

Round 1

Reviewer 1 Report

The paper from Bontoux et al. reviewing the role of Her2 in NSCLC, focusing on the therapeutic approaches is well structured. However, I suggest a general review of the English form and some minor points.

 1) NSCLC is defined as “Non-small cell lung carcinoma” in the title, while is defined as “Non-small-cell lung cancer” in the abstract (line 26). I suggest uniforming this definition as “Non-small cell lung cancer”.

2) I would eliminate “Historically” at line 31, in the abstract.

3) Tite of paragraph 4. “alterations”, not “alteration” (line 141).

4) Correct “figure 1” as “Figure 1” (line 217).

5) The legend of Figure 1 needs to be edited!

6) I would edit the title o paragraph 5 (line 225) with “Detections methods for HER2 alterations in NSCLC”.

7) I suggest inverting sub-paragraphs 6.1 and 6.2.

8) Funding information is missing.

9) In the author contribution session, is better to list the author following the order of appearance of the authorship.

Author Response

Response to Reviewer 1 Comments

Point 1: NSCLC is defined as “Non-small cell lung carcinoma” in the title, while is defined as “Non-small-cell lung cancer” in the abstract (line 26). I suggest uniforming this definition as “Non-small cell lung cancer”.`

Response 1: the definition has been uniformed, accordingly

Point 2 : I would eliminate “Historically” at line 31, in the abstract.

Response 2: we have, accordingly, modified the abstract

Point 3 : Title of paragraph 4. “alterations”, not “alteration” (line 141).

Response 3: we have, accordingly, modified the title

Point 4 : Correct “figure 1” as “Figure 1” (line 217).

Response 4: we have, accordingly, corrected the figure

Point 5 : The legend of Figure 1 needs to be edited!

Response 5: we have, accordingly, edited the legend

Point 6 : I would edit the title of paragraph 5 (line 225) with “Detections methods for HER2 alterations in NSCLC”.

Response 6: we have, accordingly, edited the title

Point 7: I suggest inverting sub-paragraphs 6.1 and 6.2.

Response 7: we thank reviewer 1 for this relevant comment. However, we think it is more understandable to organize the therapeutic part starting chronologically with a short paragrah about non-selective TKI following by a denser part about selective TKI.

Point 8: Funding information is missing.

Response 8: we have, accordingly, added funding information

Point 9: In the author contribution session, is better to list the author following the order of appearance of the authorship.

Response 9: we have, accordingly, modified the author contribution session

Reviewer 2 Report

The focus of this review will be of substantial interest to researchers engaged in basic and clinical sciences. A more structured revised version with additional background information in a language accessible to non-specialist readers would substantially improve the value and impact of this article. Any revision of the manuscript should address the comments listed below.

Comments:

1.    Authors should use a publicly available KM plotter dataset and present the data for HER2 overexpression and its impact on the clinical outcome of lung cancer patients.

2.    Please provide some background on HER2 functions as a tyrosine kinase receptor growth-promoting protein and highlight any clinical significance of its expression for metastatic NSCLC. 

3.    Please add the references for every scientific statement mentioned in the review article. For example, Line 315-318 on page 9 should have a reference at the end. 

4.    Please add the relevant research article instead of referring to the review article wherever required as it takes credit from the actual scientific addition.

5.    The manuscript also needs careful copy editing.

Author Response

Response to Reviewer 2 Comments

Point 1: Authors should use a publicly available KM plotter dataset and present the data for HER2 overexpression and its impact on the clinical outcome of lung cancer patients.

Response 1: we have, accordingly, added KM plots for HER2 mutations and HER2 amplification. KM plots for HER2 overexpression were not available.

Point 2 : Please provide some background on HER2 functions as a tyrosine kinase receptor growth-promoting protein and highlight any clinical significance of its expression for metastatic NSCLC. 

Response 2: we have, accordingly, modified the manuscript

Point 3 : Please add the references for every scientific statement mentioned in the review article. For example, Line 315-318 on page 9 should have a reference at the end. 

Response 3: we have, accordingly, added the reference

Point 4 : Please add the relevant research article instead of referring to the review article wherever required as it takes credit from the actual scientific addition.

Response 4: we have, accordingly, modified or suppressed the referrences to review articles.

Point 5 : The manuscript also needs careful copy editing.

Response 5: copy editing has been made, accordingly